# Influence of MWCNT Coated Nickel on the Foaming Behavior of MWCNT Coated Nickel Reinforced AlMg4Si8 Foam by Powder Metallurgy Process

**Ferdinandus Sarjanadi Damanik * and Günther Lange ***

Group for Metallic Materials and Composite Materials, Technische Universität Ilmenau, Gustav-Kirchhoff-Str. 6, 98693 Ilmenau, Germany

* Correspondence: ferdinandus-sarjanadi.damanik@tu-ilmenau.de (F.S.D.);
guenther.lange@tu-ilmenau.de (G.L.); Tel.: +49-(0)3677-69-1980 (F.S.D.); +49-(0)3677-69-2881 (G.L.)

**Abstract:** This research studies the effect of multi-wall carbon nanotube (MWCNT) coated nickel to foaming time on the foam expansion and the distribution of pore sizes MWCNT reinforced AlMg4Si8 foam composite by powder metallurgy process. To control interface reactivity and wettability between MWCNT and the metal matrix, nickel coating is carried out on the MWCNT surface. Significantly, different foaming behavior of the MWCNT coated nickel reinforced AlMg4Si8 was studied with a foaming time variation of 8 and 9 min. Digital images generated by the imaging system are used with the MATLAB R2017a algorithm to determine the porosity of the surface and the pore area of aluminum foam efficiently. The results can have important implications for processing MWCNT coated nickel reinforced aluminum alloy composites.

**Keywords:** aluminum matrix foam composite (AMFC); MWCNT; chemical oxidation; electroless deposition nickel; powder metallurgy; expansion

---

## 1. Introduction

The first production of metallic foam was carried out by Benjamin Sosnick in 1948 [1], extensive research and application of metallic foam in many manufacturing sectors such as the automotive and aerospace industries have been carried out. Metal foams have properties such as high strength to weight ratio, high energy absorption capacity, large specific surface, high gas and liquid permeability, and low thermal conductivity [2]. Basic characteristics such as the relative density, cell structure, homogenity of cell morphology and optimum mean average equivalent diameter of pores are things that affect the use of metal foams [3]. Some parameters that affect the quality of the final product foam are: particles (composition, shape, size and volume fraction), gas (composition and purity), particle-surface interaction, matrix alloy composition, foaming temperature and thermal processing conditions (holding time and cooling medium) [4].

In aluminum foam manufacturing processes, commonly known are direct gas foaming, powder metallurgy and casting [5]. Some researchers have investigated adding ceramic particles to aluminum foam, but the main problem for ceramic reinforced aluminum foam is agglomeration and poor bonding between the reinforcement and the matrix, which sometimes results in brittleness deformation and decreased energy absorption efficiency of foam composites [6]. Therefore, it is necessary to develop the type of the new favorable reinforcement and surface treatment of the reinforcement, specifically for the nano-sized reinforcement and to exploit the properties of composite foam. Research of nano and microparticles reinforced closed-cell aluminum foam composites by the powder metallurgy method has been investigated. Compressive strength and the absorption of plastic deformation energy increase

about 28 times with the addition of nano-SiC particles when compared to micro-SiC reinforcing aluminum foam [7].

The strengthening effects of $Cr_2O_3$ and $Al_3Ni$ thermally grown on an aluminum alloy were investigated, the strengths and stiffness of aluminum were enhanced. $Al_3Ni$ layers are also formed on graphite coated nickel reinforced aluminum. The primary $Al_3Ni$ intermetallic formed in the aluminum and nickel reactions is the angular $Al_3Ni$ phase. Layer morphology is an important role in the interface and mechanical properties of composites. The $Al_3Ni$ phase plays a role in binding carbon in the aluminum matrix. These results indicate that the dispersion of graphite in the aluminum matrix is facilitated by a layer of nickel in graphite [8,9]. MWCNT has the potential to be used as a reinforcement in composite materials because MWCNT has superior mechanical properties with tensile strengths of up to 150 GPa and elastic modulus up to 1 TPa, as well as good thermal stability and electrical conductivity [10]. The effect of MWCNT volume fraction (0.1–1.0 wt.%) has been investigated on the compressive behavior and energy absorption of nanocomposite foams. Compressive yield strength increased with the addition of up to 0.5 wt.% MWCNT, but the opposite trend occurred with the addition of 1 wt.% MWCNT [11]. To improve dispersion and wettability and overcome agglomeration and poor bonding problems between MWCNT and aluminum matrix, it can be done with MWCNT surface treatment (chemical oxidation and electroless deposition nickel coating). Chemical oxidation of MWCNT involves impregnation of functional groups (functional groups -COOH) on the surface of nanotubes because the reactivity of oxygen-containing groups is greater than pure carbon [12]. MWCNT oxidation (MWCNT-COOH) can trap much more Sn and Pd following nickel nucleation sites and will help to incorporate MWCNT into a metal matrix with strong bonding. Ni ions easily accept electrons and electrodeposit selectively on the defect sites rather than on other normal sites on the outer surface of MWCNT [13]. The effect of surface treatment on MWCNT (chemical oxidation followed by nickel coating on MWCNT) reinforced AlMg4Si8 by powder metallurgy has been studied, showing uniformly distribution and good bonding on MWCNT reinforced aluminum foam [14].

The goal of this the present research is to investigate the effect of MWCNT coated Nickel to foaming time on the foam expansion and the distribution of pore sizes of the MWCNT reinforced AlMg4Si8 foam composite (AMFC-aluminum matrix foam composite) by the powder metallurgy method. Determination of surface porosity, pore area and foam shape of the aluminum foam was obtained through an imaging system with the MATLAB R2017a algorithm.

## 2. Materials and Methods

### 2.1. Materials

Aluminum alloy AlMg4Si8 was used for this study and the average particle size and purity of the metal, MWCNT and foaming agent powders can be seen in Table 1. Figure 1 show field emission scanning electron microscopes (FE-SEM) image morphology of aluminum, magnesium, silicon, $TiH_2$ and MWCNT in this study.

**Table 1.** Characteristics of powders used as starting materials.

| Powder | Supplier | Purity | Size |
|--------|----------|--------|------|
| Al | TLS Technik (Bitterfeld, Germany) | 99.7 | <150 μm |
| Mg | Laborladen.de (Hüfingen, Germany) | 99.8 | <40 μm |
| Si | - | >99.95 | <40 μm |
| $TiH_2$ | Alfa Aesar | 99 | <45 μm |
| MWCNT | Sigma Aldrich | >90 | 5–9 μm |

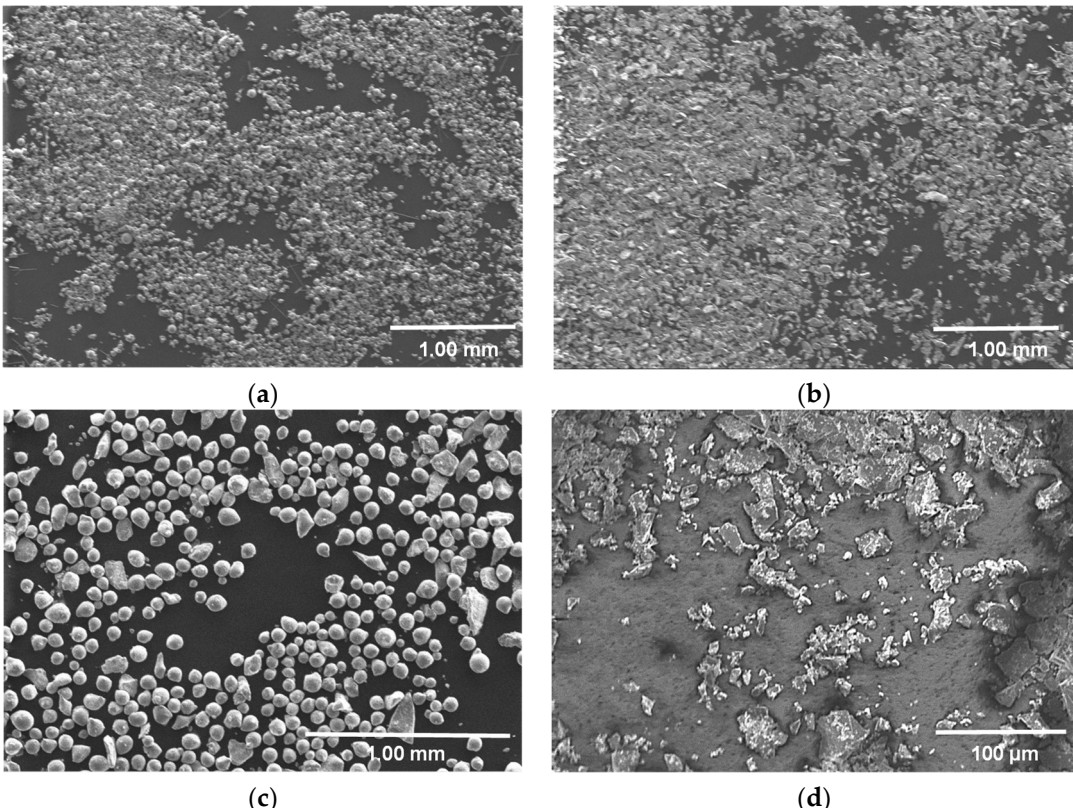

**Figure 1.** Field emission scanning electron microscopes micrographs of (**a**) Spheroidal powder particles Al; (**b**) Angular powder particles Mg; (**c**) Spheroidal powder particles Si; (**d**) Angular powder particles TiH$_2$ (TU Ilmenau; FG MWV).

## 2.2. Method

MWCNT were oxidized in piranha reagent (3 H$_2$SO$_4$: 1 hydrogen peroxide) for 6 h. After the treatments, all the samples were thoroughly washed with demineralized water [15]. After this treatment, the procedure of electroless plating nickel can be divided into three steps: sensitization, activation and plating. MWCNT-oxidized were immersed in a sensitization solution and sonicated for 30 min. The Sn$^{2+}$-sensitized MWCNT were further stirred in an activating solution for 30 min. The activated MWCNT were washed with deionized water and introduced into an electroless plating bath. After 10 min of plating, the plated MWCNT were washed with distilled water then with methanol [16].

Aluminum closed-cell foams were prepared using by powder metallurgy process. Before mixing, MWCNT coated nickel were sonicated with ethanol in a bath sonicator for 30 min. The process started with the mixing of appropriate amounts of basic ingredients, AlMg4Si8 powder, MWCNT (0.5 wt.%) and TiH$_2$ (0.5 wt.%) as blowing agents, inside a v-shaped cylinder for 20 min. AlMg4Si8, 0.5 wt.% TiH$_2$ and 0.5 wt.% MWCNT were prepared inside a stainless steel mold, then axially compacted (cold pressing) with the pressure of 450 MPa for 3 min. It was heated in a preheated furnace at 300 °C for 2 h. The precursor (volume: 19.2325 cm$^3$) was foamed at 750 °C for 8 min. FE-SEM techniques were employed to characterize the samples to examine the dispersion of the MWCNT within the AlMg4Si8 foam matrix [17]. The sample was investigated using inexpensive digital systems/devices camera. In this study, the following steps show how the images were acquired, processed and calculated the area of each pore on the camera digital and measured the pores size, shape and size distribution using MATLAB R2017a (with appropriate MATLAB commands added). Flowchart (Figure 2) outlines the algorithm of the image processing and analyses executed in the MATLAB program for images obtained with each imaging system [18].

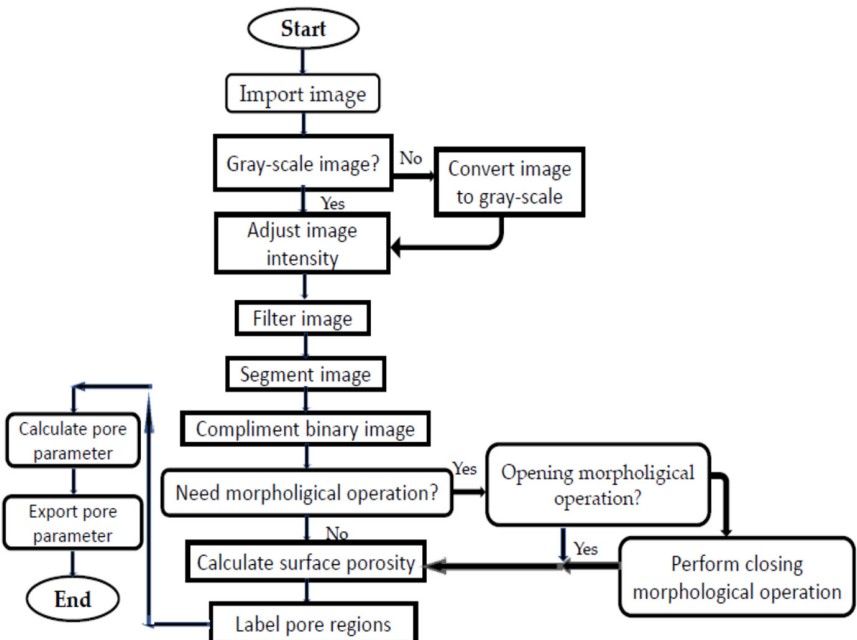

**Figure 2.** Algorithm of image processing and analyses executed in MATLAB R2017a Reproduced from [18], with permission from Polymers, 2019.

## 3. Result

*3.1. The Effect of Multi-Wall Carbon Nanotube Coated Nickel to Foaming Time on Morphology of AlMg4Si8 Foam*

Figure 3 shows the images of produced samples using different reinforcement and foaming time, macroscopic images of aluminum foam were taken using a camera. The results demonstrate that the effect of surface treatment of the MWCNT influence the expansion and microstructure of AlMg4Si8 foam nanocomposite and the cell number density decrease with the addition of reinforcement MWCNT coated nickel in AlMg4Si8. The foam expansion of AlMg4Si8 foam without MWCNT has the highest expansion and shows relatively uniform cells compared to MWCNT coated nickel reinforced AlMg4Si8. The results show the effect of MWCNT coated nickel to foaming time on the foamability of MWCNT reinforced aluminum foam. MWCNT reinforced aluminum foam for 8 min show a smaller size of pore compared to MWCNT reinforced aluminum foam for 9 min and the volumetric expansion coefficient of MWCNT reinforced aluminum foam for 8 min (c) is 0.00415 1/°C, lower than the MWCNT reinforced aluminum foam for 9 min (d), which is 0.00623 1/°C.

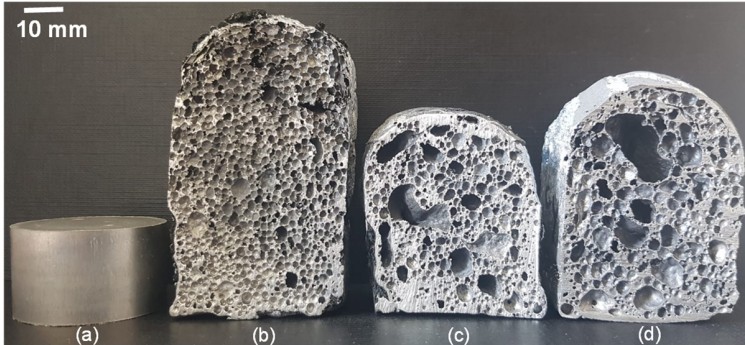

**Figure 3.** Macrostructure of aluminum foam expansion affected by reinforcement and foaming time of (**a**) Precursor material Multi-wall Carbon Nanotube reinforced AlMg4Si8; (**b**) AlMg4Si8 foam; (**c**) MWCNT coated nickel reinforced AlMg4Si8 foam for 8 min; (**d**) MWCNT coated nickel reinforced AlMg4Si8 foam for 9 min. (TU Ilmenau; FG MWV).

Figure 4 shows that the distribution of MWCNT coated nickel in aluminum foam are uniform on the cell wall. FE-SEM analysis results show that all reinforcement (MWCNT) are evenly distributed in the aluminum foam matrix (the top, middle and bottom layers). The presence of MWCNT coated nickel is in the cell wall which means good the wettability at the MWCNT coated nickel reinforced aluminum foam interface.

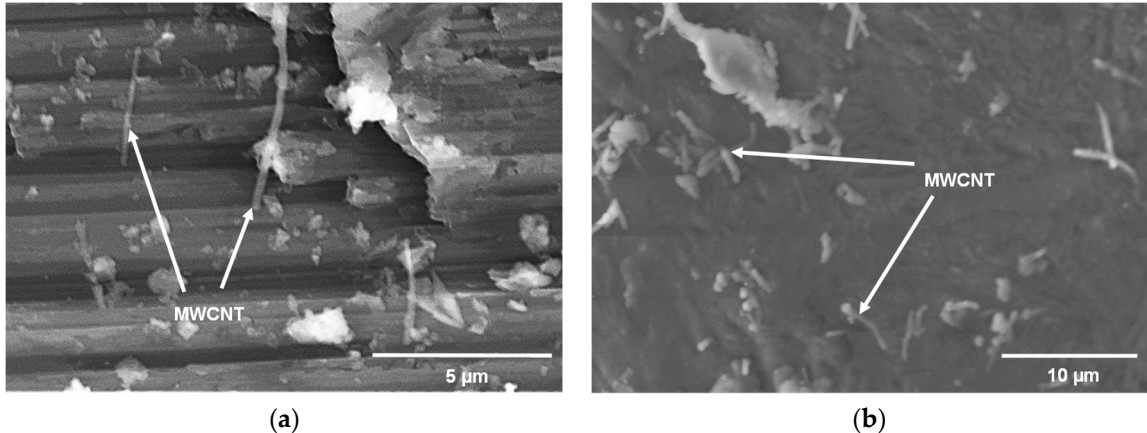

(**a**)                  (**b**)

**Figure 4.** (**a**) Field emission scanning electron microscopes image of dispersion MWCNT on the plateau border; (**b**) Show microstructural observed by field emission scanning electron microscopes about the uniform dispersion of MWCNT in the pore cell-wall. (TU Ilmenau; FG MWV).

Table 2 shows foam density, % porosity, expansion and pore number of foam. The porosity of foam material were calculated using the following equation:

$$\% \text{ Porosity} = 1 - \frac{\rho \text{ foam}}{\rho \text{ precursor}} \times 100\% \tag{1}$$

The density of precursors and foam were measured by weighing specimens and geometry, that all the calculations are in comparison with the bulk aluminum, where $\rho1$ and $\rho2$ are densities of foam and precursor. The volumetric expansion coefficient was measured by volume foam compare with the precursor, where $\alpha$ = volumetric thermal expansion coefficient, V0 = the initial volume, $\Delta$V = volume change and $\Delta$T = temperature change. Furthermore, the relative density of precursors and foams and the volumetric expansion coefficient were calculated using the following equations:

$$\rho R = \frac{\rho1}{\rho2} \tag{2}$$

$$\alpha = \frac{\Delta V}{V0.\Delta T} \tag{3}$$

**Table 2.** Structure of the produced aluminum foam.

| Foam Sample | Time (min) | Density (g/cm$^3$) | Porosity (%) | The Volumetric Expansion Coefficient (1/°C) | Pore Number |
|---|---|---|---|---|---|
| AlMg4Si8 (a) | 8 | 0.5573 | 72.32 | 0.00693 | 2605 |
| MWCNT(Ni)-AlMg4Si8 (b) | 8 | 0.8368 | 67.81 | 0.00415 | 726 |
| MWCNT(Ni)-AlMg4Si8 (c) | 9 | 0.5581 | 78.55 | 0.00623 | 484 |

### 3.2. The Pore Size Distribution of Multi-Wall Carbon Nanotube Coated Nickel Reinforced AlMg4Si8

To compare the porosity of the surface and the pore area of aluminum foam are used the MATLAB algorithm from digital images produced by the imaging system shown in Figure 5. The imaging system shows a plot diagram of the frequency distribution bar of the pore area, the number of large pores relative to the small pores in the image by each imaging system. A summary of the number of pores detected in all analyzed samples affected by the reinforcement and foaming time is shown in Table 2. The structure of AlMg4Si8 without reinforcement show relatively uniform cells than the MWCNT coated nickel reinforced AlMg4Si8 and imaging systems show an inhomogeneous distribution of large and small pores throughout the aluminum foam, and this is caused by the influence of the type of MWCNT as a reinforcement and foaming time used in the foaming process/manufacturing.

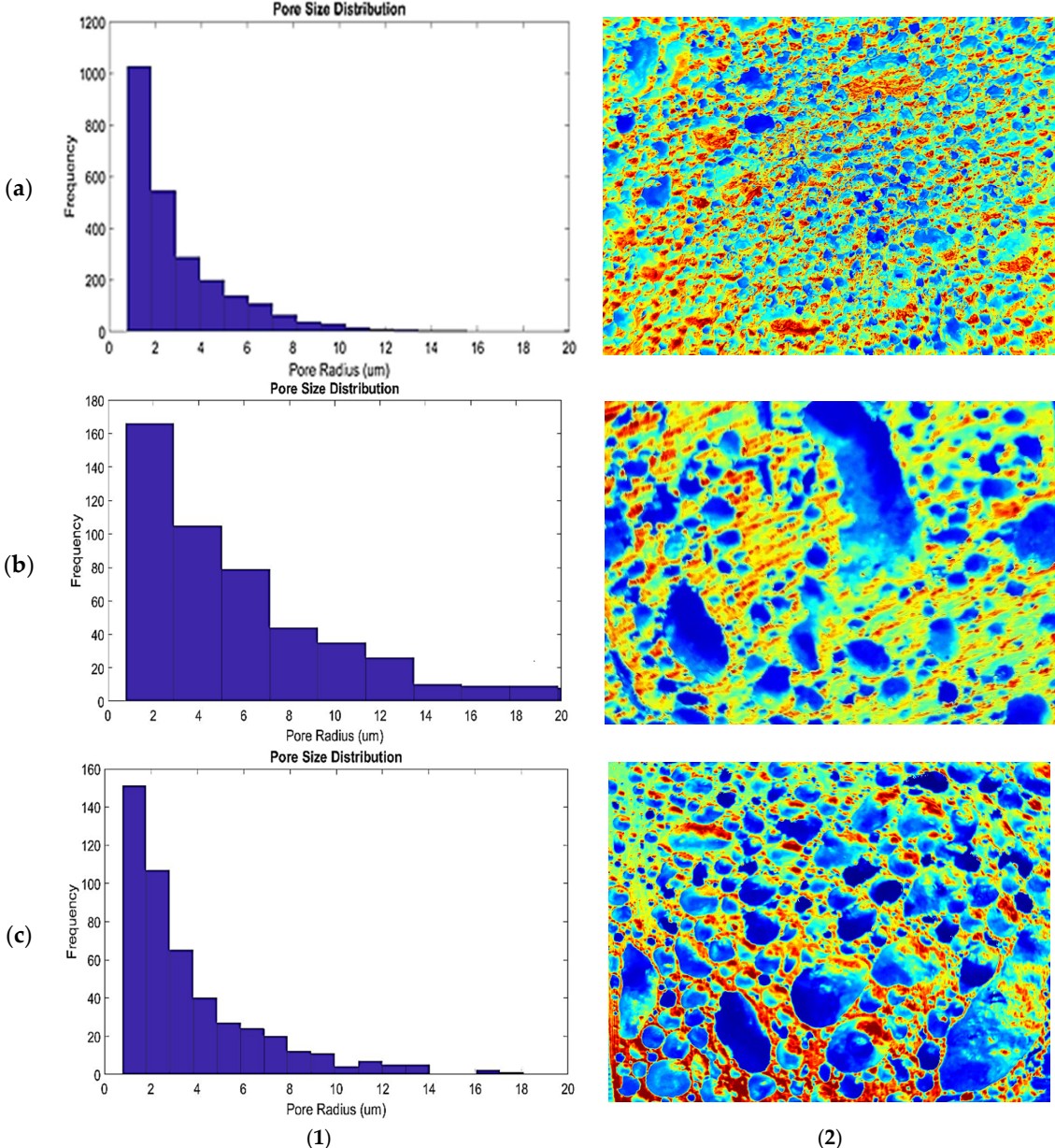

**Figure 5.** Frequency distribution of pore area (**1**), and Scaled labeled images (**2**) for (**a**) AlMg4Si8 foam for 8 min (**b**) MWCNT coated nickel reinforced AlMg4Si8 foam for 8 min (**c**) MWCNT coated nickel reinforced AlMg4Si8 foam for 9 min.

## 4. Discussion

Several studies on the effect of pore size on the mechanical properties of the aluminum foam and its alloys have found that porosity and pore size distribution have a completely linear relationship on mechanical properties. Furthermore, the fabrication method and composition used during foam making will affect the foam structure and pore morphology, which will also determine the mechanical properties [19,20].

To discuss the effect of reinforcement MWCNT coated nickel and foaming time on expansion and pore size has to take into account the following aspect. Chemical oxidation and electroless deposition nickel in MWCNT can improve the distribution and wettability of MWCNT as reinforcement in an aluminum foam nanocomposite. Nickel coating on MWCNT can increase the bond between MWCNT coated nickel and aluminum matrix. While MWCNT coated nickel were much more strongly bonded with the aluminum matrix. Figure 4 shows that the MWCNT coated nickel is in a foam cell and has better wettability with the aluminum foam matrix. This investigation shows that the MWCNT surface structure affects the interface between MWCNT and aluminum matrix in the aluminum foam matrix nanocomposite.

Digital images produced by the imaging system are used with the MATLAB algorithm to determine the surface porosity and pore area of aluminum foam. The average diameter of the pores MWCNT coated nickel reinforced AlMg4Si8 increase than AlMg4Si8 without reinforcement. The porosity, the volumetric expansion coefficient and density of all samples are listed in Table 2. According to Table 2, samples a, b and c showed the porosity of each sample (72.32, 67.81, and 78.55%, respectively). The volumetric expansion coefficient (0.00693, 0.00415 and 0.00623 1/°C, respectively) and density (0.5573, 0.8368 and 0.5581 g/cm$^3$, respectively). Increasing the foaming time MWCNT coated nickel reinforced AlMg4Si8 from 8 to 9 min resulted in a rapid increase of the volumetric expansion coefficient (0.00415 to 0.00623 1/°C) and consequently a rapid decrease in density (0.8368 to 0.5581 g/cm$^3$). Table 2 shows at the same time for 8 min, the volumetric expansion coefficient of AlMg4Si8 (without MWCNT) is 0.00693 1/°C, while the volumetric expansion coefficient of the MWCNT coated nickel reinforced AlMg4Si8 is 0.00415 1/°C. Increasing the foaming time from 8 to 9 min resulted in an increase of the volumetric expansion coefficient (0.00415 to 0.00623 1/°C), and consequently a rapid decrease of the relative density (0.8368 to 0.5581 g/cm$^3$).

According to data from Table 2, the comparison of the structure of the aluminum foam produced are:

Pores Number: AlMg4Si8 (8 min) > MWCNT(Ni)–AlMg4Si8(8 min) > MWCNT(Ni)–AlMg4Si8 (9 min)

% Porosity: MWCNT(Ni)–AlMg4Si8 (8 min) < AlMg4Si88 (8 min) < MWCNT(Ni)–AlMg4Si8 (9 min)

Pore size: AlMg4Si8 (8 min) < MWCNT(Ni)–AlMg4Si8(8 min) < MWCNT(Ni)–AlMg4Si8(9 min)

Density: AlMg4Si8 (8 min) < MWCNT(Ni)–AlMg4Si8(9 min) < MWCNT(Ni)–AlMg4Si8(8 min)

The volumetric expansion coefficient: AlMg4Si8 (8 min) > MWCNT(Ni)–AlMg4Si8 (9 min) > MWCNT(Ni)–AlMg4Si8 (8 min)

The results show that the effect of MWCNT surface treatments as reinforcement of the foamability of AlMg4Si8. During the processing of aluminum foam composites, chemical reactions occur at the interface between the matrix and the reinforcement in several systems. Chemical reactions and the types of reaction products formed depend on the processing temperature, pressure and atmosphere, matrix composition and surface chemistry of the reinforcement. The physical and mechanical properties of aluminum foam composites are greatly influenced by the degree of the chemical reaction. The inhomogeneous distribution of large and small pores in all-aluminum foams is shown by the aluminum foam composite imaging system (Figure 5), and this is influenced by MWCNT coated nickel as the reinforcement used and the foaming time in the manufacturing process. The structure of AlMg4Si8 without reinforcement show cells that are relatively uniform compared to MWCNT coated nickel reinforced AlMg4Si8. This shows that MWCNT coated nickel is collected inside the cell wall during foaming and there has been the wettability between MWCNT and aluminum matrix. When aluminum wets the MWCNT coated nickel, the presence of MWCNT can increase the system viscosity

and will increase the cell wall thinning rate and as consequence, larger cells will form in aluminum foam and reduce the number of pores.

## 5. Conclusion

AlMg4Si8 foam is produced by a powder metallurgical process in which titanium hydride is a blowing agent. By varying foaming time and compare with AlMg4Si8 foam (without MWCNT), different stages of early foam formation could be prepared. Influence of MWCNT coated nickel to the foaming time and distribution of pore on the foaming behavior of MWCNT coated nickel reinforced aluminum alloys and the influence of time has been investigated, that:

Uniformly dispersion of MWCNT coated nickel in the aluminum matrix and increase the interfacial bonding strength between MWCNT coated nickel and metal matrix.

Surface treatment on MWCNT will increase the cell wall thinning rate and as a consequence, larger cells will form in aluminum foam. The average diameter of the pores MWCNT coated nickel reinforced AlMg4Si8 is greater than AlMg4Si8 foam.

MWCNT coated nickel reinforced AlMg4Si8 with time for 9 min has a higher volumetric expansion coefficient (0.00623 1/°C) than the foaming time for 8 min (0.00415 1/°C). However, the volumetric expansion coefficient of AlMg4Si8 without reinforcement with a foaming time of 8 minutes still has the highest volumetric expansion coefficient value (0.00693 1/°C).

**Author Contributions:** Conceptualization, F.S.D. and G.L.; methodology, F.S.D. and G.L.; validation, F.S.D. and G.L.; software, F.S.D.; formal analysis, F.S.D.; investigation, F.S.D; data curation, F.S.D.; writing—original draft preparation, F.S.D.; writing—review and editing, F.S.D. and G.L.; visualization, F.S.D. and G.L; supervision, G.L. All authors have read and agreed to the published version of the manuscript.

**Funding:** This work is supported and funded by Indonesia Endowment Fund for Education (LPDP-Lembaga Pengelola Dana Pendidikan).

**Acknowledgments:** SEM analysis facilities were made accessible by The Institute of Micro- and Nanotechnologies Technology center TU Ilmenau. We thank our colleagues from The Department of Metal and Composite Materials Technical University Ilmenau, who provided powders and contributed to the discussions that greatly assisted the research.

**Conflicts of Interest:** The authors declare no conflict of interest. The funders had no role in the design of the study; in the collection, analyses, or interpretation of data; in the writing of the manuscript; or in the decision to publish the results.

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
