# Peer review of "Influence of MWCNT Coated Nickel on the Foaming Behavior of MWCNT Coated Nickel Reinforced AlMg4Si8 Foam by Powder Metallurgy Process"

_metals, doi:10.3390/met10070955_

Round 1

Reviewer 1 Report

Lines 108-109. Based on Figure 3, it is difficult to determine if the number of pores shown in Figure 3d and Figure 3c is significantly different. In addition, the expansion is twice as large compared to what?

In Figure 4 there are no microphotographs 4c, d, listed in the signature.

The formula on page 5 and the first formula on page 6 have the same sign (1).

Table 2 shows Linear Exp [cm], not aLE [%].

Why sample (a) shown in Table 2, despite the larger Linear Expansion has lower porosity than sample (c)?

The description of the axes in Figure 5 is completely illegible.

Lines 150-154. On what basis is this statement? Literature?

Lines 155-165. These are the suppositions. There is no evidence in the submitted documentation for such statements.

Line 171 The expansion dimension does not comply with Table 2. In addition, when comparing the heights of the samples shown in Figure 3 and substituting them for the formula (2), the expansion should be for sample (a) - 133%, for sample (b) -92% and for sample (c) - 117%.

Lines 218, 221. Two references have the same numbering.

The work is edited carelessly and is not suitable for publication in this version.

Reviewer 2 Report

Review Report: metals-836284

The manuscript, # metals-836284, entitled “Influence of MWCNT Coated Nickel to The Foaming Time and Distribution of Pore on the Foaming Behavior of MWCNT Coated Nickel Reinforced AlMg4Si8 Foam by Powder Metallurgy Process” written by Ferdinandus Damanik and G. Lange, reported the effect of MWCNT coated nickel to foaming time on the foam expansion and the distribution and has the worth of publication in Metals. This manuscript is well organized and giving the good information to the technical and scientific community. This manuscript should be acceptable after removing some writing errors as given below:

  1. Introduction, Authors have to cite some references regarding Ni-alloys where, TGO like Cr2O3 and Al2O3 have improved the mechanical strength. Authors have to explain why they need to coat MWCNT on Ni, even though, TGO work well as the mechanical strength and cite the suggested references [Journal of Nuclear Materials 389 (2009) 420–426, Journal of Nuclear Materials 405 (2010) 165–170, Journal of the European Ceramic Society 29 (2009) 355–362].
  2. There are some grammatical errors what they have to improve before publication.

Reviewer 3 Report

Very interesting study for exploring  the effect of nickel coating on MWCNT  and  foaming time,  on linear  expansion and the distribution of pore sizes on the MWCNT reinforced AlMg4Si8 foam composites. By using experimental imaging and Matlab,  the authors have shown the surface porosity, foam structure and the distribution of pores within  the MWCNT coated nickel reinforced AlMg4Si8 materials. Surface porosity and the pore distribution can have important implications in the mechanical properties of foaming materials and this study will add important insight  in this area. However, this paper is hard to read and needs a round of English editing. I have some comments and corrections which are shown below.

  1. Simplify the Title of the paper. It is difficult to understand what it means
  2. provide the full name of MWCNT before abbreviating
  3. line 32: it seems the sentence is not complete
  4. give scale bar for Figure 1 and 4
  5. line 118-121 is hard to follow
  6. Figure 5: the font size used in the X and Y axis of the frequency distribution plots are too small to read.
  7. line 155: there is a typo “efect”
  8. line 169-172: the line start with “ according to table 2……” a,b,c show highest porosity compare to what?
  9. the text between line 178-179 is not very clearly described. what does the % refer to ?
  10. The explanation of why there are large pores form  on the MWCNT coated nickel reinforced AlMg4Si8 foam is repetitively used  in the text. e.g 190-193 and 160-163
  11. This is very interesting that the foaming time of 8 min versus 9 minutes has such a big difference in the pore size distribution and on the linear expansion. Did you consider experiments for different time scale besides 8 and 9 minutes?
  12. Does Figure 3 b,c,d correspond to Figure 5 a,b,c? The linear expansion reported in the table 2 is strange. Just looking by eye sample c (25%) versus d (40%) in Figure 3 does not look that different.

Round 2

Reviewer 1 Report

Thank you for the explanations. They satisfy me for the most part. However, I propose to harmonize the designations in formula (3) with index 1 for precursor and index 2 for foam.
However, I cannot agree with answer 5. After all, it is physically impossible to obtain a greater porosity of the sample and at the same time a higher density as shown in Table 2 for samples (a) and (c). Please analyze the attached calculations.

Round 3

Reviewer 1 Report

If the precursor has the form of a cylinder and the foams are cuboids, i.e. solids of a different shape, then the analysis of the linear expansion coefficient makes no sense. In order to correctly analyze the expansion, one should determine the volumetric expansion coefficient. Please make such a change in table 2 and in the “discussion” chapter.
